

# An empirical Bayes approach to stochastic blockmodels and graphons: shrinkage estimation and model selection

Zhanhao Peng and Qing Zhou

Department of Statistics, University of California, Los Angeles, Los Angeles, California, United States of America

## ABSTRACT

The graphon (W-graph), including the stochastic block model as a special case, has been widely used in modeling and analyzing network data. Estimation of the graphon function has gained a lot of recent research interests. Most existing works focus on inference in the latent space of the model, while adopting simple maximum likelihood or Bayesian estimates for the graphon or connectivity parameters given the identified latent variables. In this work, we propose a hierarchical model and develop a novel empirical Bayes estimate of the connectivity matrix of a stochastic block model to approximate the graphon function. Based on our hierarchical model, we further introduce a new model selection criterion for choosing the number of communities. Numerical results on extensive simulations and two well-annotated social networks demonstrate the superiority of our approach in terms of parameter estimation and model selection.

## INTRODUCTION

Network data, consisting of relations among a set of individuals, are usually modeled by a random graph. Each individual corresponds to a vertex or node in the graph, while their relations are modeled by edges between the vertices. Such data have become popular in many domains, including biology, sociology and communication (*Albert & Barabási, 2002*). Statistical methods are often used to analyze network data so that the underlying properties of the network structure can be better understood *via* estimation of model parameters. Examples of such properties include degrees, clusters and diameter among others (*Barabási & Albert, 1999*; *Newman, Watts & Strogatz, 2002*).

To better understand the heterogeneity among vertices in a network, community detection and graph clustering methods (*Girvan & Newman, 2002*; *Newman, 2004*) have been proposed to group vertices into clusters that share similar connection profiles. A large portion of the clustering methods are developed based on the stochastic block model (SBM) (*Freeman, 1983*), which constructs an interpretable probabilistic model for the heterogeneity among nodes and edges in an observed network.

For an undirected simple random graph on $n$ nodes or vertices, the relationships between the nodes are modeled by $\frac{1}{2}n(n-1)$ binary random variables representing the presence or absence of an undirected edge. The edge variables can be equivalently

Corresponding author
Qing Zhou, zhou@stat.ucla.edu

represented by an $n \times n$ adjacency matrix $\mathbf{X}$, where $X_{ij} = 1$ if node $i$ and $j$ are connected and $X_{ij} = 0$ otherwise. We do not consider self loops in this work, and thus $X_{ii} = 0$ for $i = 1, \ldots, n$.

Many popular graph models (*Lloyd et al., 2012*) make exchangeability assumption on the vertices: The distribution of the random graph is invariant to permutation or relabeling of the vertices. A large class of exchangeable graphs can be defined by the so-called *graphon* function (*Lovasz & Szegedy, 2006*). A graphon $W(u, v)$ is a symmetric function: $[0, 1]^2 \rightarrow [0, 1]$. To generate an $n$-vertex random graph given a graphon $W(u, v)$, we first draw latent variables $u_i$ independently from the uniform distribution $\mathrm{U}(0, 1)$ for $i = 1, \ldots, n$. Then we connect each pair of vertices $(i, j)$ with probability $W(u_i, u_j)$, i.e.,

$$\mathbb{P}(X_{ij} = 1 | u_i, u_j) = W(u_i, u_j), \quad i, j = 1, \ldots, n. \tag{1}$$

In particular, the stochastic block model mentioned above can be seen as a special case of the graphon model, where $W(u, v)$ is a piecewise constant function. *Abbe (2018)* has summarized recent developments on the stochastic block model. Under an SBM, the vertices are randomly labeled with independent latent variables $\mathbf{Z} = (z_1, \ldots, z_n)$, where $z_i \in \{1, \ldots, K\}$ for $i = 1, \ldots, n$ and $K$ is the number of communities or clusters among all the nodes. The distribution of $(\mathbf{Z}, \mathbf{X})$ is specified as follows:

$$
\begin{aligned}
\mathbb{P}(z_i = m) &= \pi_m, \quad m \in \{1, \ldots, K\}, i = 1, \ldots, n, \\
\mathbb{P}(X_{ij} = 1 | z_i, z_j) &= \theta_{z_i z_j}, \quad i, j = 1, \ldots, n,
\end{aligned}
\tag{2}
$$

where $\sum_m \pi_m = 1$ and each $\theta_{km} \in [0, 1]$. Put $\pi = (\pi_1, \ldots, \pi_m)$ and $\Theta = (\theta_{ij})_{K \times K}$.

Many efforts have been made on statistical inference of the SBM to detect block structures as well as to estimate the connectivity probabilities in the blocks. Some classical and popular methods include MCMC, degree-based algorithms and variational inference among other. *Nowicki & Snijders (2001)* developed a Gibbs sampler to estimate parameters for graphs of small sizes (up to a few hundred nodes). A degree-based algorithm (*Channarond, Daudin & Robin, 2012*) achieves classification, estimation and model selection from empirical degree data. The variational EM algorithm (*Daudin, Picard & Robin, 2008*) and variational Bayes EM (*Latouche, Birmele & Ambroise, 2012*) approximate the conditional distribution of group labels given the network data by a class of distributions with simpler forms. *Suwan et al. (2016)* recast the SBM to a random dot product graph (*Young & Scheinerman, 2007*) and developed a Bayesian inference method with a prior specified empirically by adjacency spectral embedding.

Due to higher model complexity, estimating a graphon is challenging. Some works (*Airoldi, Costa & Chan, 2013*; *Olhede & Wolfe, 2014*; *Latouche & Robin, 2016*) have focused on the nonparametric perspective of this model and developed methods to estimate a graphon based on SBM approximation. These methods estimate a graphon function by partitioning vertices and computing the empirical frequency of edges across different blocks. Many algorithms put emphasis on model selection (*Airoldi, Costa & Chan, 2013*) or bandwidth determination (*Olhede & Wolfe, 2014*). *Latouche & Robin*

*(2016)* proposed a variational Bayes approach to graphon estimation and used model averaging to generate a smooth estimate.

Meanwhile, model selection that compares different node clustering schemes and selects the most appropriate number of blocks for SBMs has been one of the major difficulties in this field. Methods that are generally applicable to all graph clustering results include a hypothesis testing based method for SBMs (*Côme & Latouche, 2015*) and a cross-validation scheme for graphons (*Airoldi, Costa & Chan, 2013*). *Côme & Latouche (2015)* propose an exact integrated complete data likelihood criterion that is combined with a greedy inference algorithm to identify node clusters for SBMs. *Yang et al. (2021)* summarize different model selection methods for spectral graph clustering and propose a simultaneous model selection framework.

After the block structure of a network is identified, most of the above methods simply use the empirical connection probability within and between blocks to estimate $\Theta$. When the number of nodes in a block is too small, the estimate can be highly inaccurate with a large variance. *Latouche & Robin (2016)* developed an alternative method under a Bayesian framework, where they put conjugate priors on the parameters $(\pi, \Theta)$. In particular, they assume $\theta_{ab} \sim \text{Beta}(\alpha_{ab}, \beta_{ab})$ independently for $a, b \in \{1, \ldots, K\}$, where the parameters $(\alpha_{ab}, \beta_{ab})$ in the prior are chosen *in priori*. Similar to the MLE, the connection probability $\theta_{ab}$ of each block is estimated separately and thus may suffer from the same high variance issue for blocks with a smaller number of nodes. To alleviate this difficulty, we propose a hierarchical model for network data to borrow information across different blocks. Under this model, we develop an empirical Bayes estimator for $\Theta = (\theta_{ab})$ and a model selection criterion for choosing the number of blocks. Empirical Bayes method is usually seen to have better performance when estimating many similar and variable quantities (*Efron, 2010*). This inspires our proposal as the connection probabilities can be similar across many different communities. By combining data from many blocks, estimates will be much more stable even if the number of nodes is small (as small as a few nodes) in each block.

In summary, our method has two major novel components: (1) shrinkage estimation for connectivity parameters, and (2) a novel likelihood-based model selection criterion, both under our proposed hierarchical model. As demonstrated by extensive simulations and experiments on real-world data, these contributions give us substantial gain in estimation accuracy and model selection performance, especially for graphons. Moreover, our method is very easy to implement and does not cost much extra computational resources compared to existing approaches.

The article is organized as follows. First, we will develop our empirical Bayes method for the SBM and the graphon, focusing on connection probability estimation and model selection on the number of blocks. Then we will compare the performance of our methods with other existing methods on simulated data and on two real-world networks. The article is concluded with a brief discussion. Some technical details and additional numerical results are provided in the Supplemental Material.

## METHODS

Let us first consider the SBM. After the vertices of an observed network have been partitioned into clusters by a graph clustering algorithm, we develop an empirical Bayes estimate of the connection probability matrix $\Theta$ based on a hierarchical binomial model. Under this framework, we further propose a model selection criterion to choose the number of blocks. Our method consists of three steps:

- **Graph clustering** For a network with $n$ vertices, cluster the vertices into $K$ blocks by a clustering algorithm. Let $Z : [n] \to [K]$ denote the cluster assignment, where $[m] := \{1, \ldots, m\}$ for an integer $m$.
- **Parameter estimation** Given $Z$, we find an empirical Bayes estimate $\widehat{\Theta}_{\text{EB}} = (\hat{\theta}_{ij}^{\text{EB}})_{K \times K}$ by estimating the hyperparameters of the hierarchical binomial model.
- **Model Selection** Among multiple choices of $K$, we select the $\hat{K}$ that maximizes a penalized marginal likelihood under our hierarchical model.

We will also generalize our method to the graphon model, following the idea of SBM approximation to a graphon.

Algorithms to detect blocks of a stochastic block model have been widely studied, including spectral clustering by *Rohe, Chatterjee & Yu (2011)*, Monte Carlo sampling by *Nowicki & Snijders (2001)* and variational approximations by *Daudin, Picard & Robin (2008)*. As an extension to the work of *Daudin, Picard & Robin (2008)*, *Latouche, Birmele & Ambroise (2012)* proposed a variational Bayes approximation to the posterior distribution of the parameters $(\pi, \Theta)$ and of the latent cluster labels **Z** (see Supplemental Material for a more detailed review). Given the **Z** estimated by their approach, we will develop our hierarchical model and empirical Bayes estimates.

### Estimating connection probabilities

In this subsection, we consider the SBM and assume a partition $Z : [n] \to [K]$ of the nodes is given, where $K$ is the number of blocks. Note that $Z^{-1}(a)$ for $a \in [K]$ is the subset of nodes in the $a$-th cluster. Let

$$B_{ab} = \{(i, j) : (i, j) \in Z^{-1}(a) \times Z^{-1}(b), i < j\}$$

be the collection of node pairs in the $(i, j)$th block. According to the SBM, the connection probability between any $(i, j) \in B_{ab}$ is $\theta_{ab}$. Recall that $\mathbf{X} = (X_{ij})$ is the observed adjacency matrix. Let $X_{ab}^B = \sum_{(i,j) \in B_{ab}} X_{ij}$ be the number of edges in block $(a, b)$. Then, we have

$$X_{ab}^B | \theta_{ab} \sim \text{Binomial}(n_{ab}, \theta_{ab}), \tag{3}$$

where $n_{ab} = |B_{ab}| = |Z^{-1}(a)| \cdot |Z^{-1}(b)|$ for $a \neq b$ and $n_{aa} = |Z^{-1}(a)| \cdot (|Z^{-1}(a)| - 1)/2$ as self loops are not allowed. Based on the empirical frequency of edges in the block $(a, b)$, we have an MLE for the edge connection probability

$$\hat{\theta}_{ab}^{\text{MLE}} = \frac{X_{ab}^B}{n_{ab}}, \qquad a, b \in \{1, \ldots, K\}. \tag{4}$$

When $K$ is large, the number of nodes, and thus $n_{ab}$, in some blocks will be small,

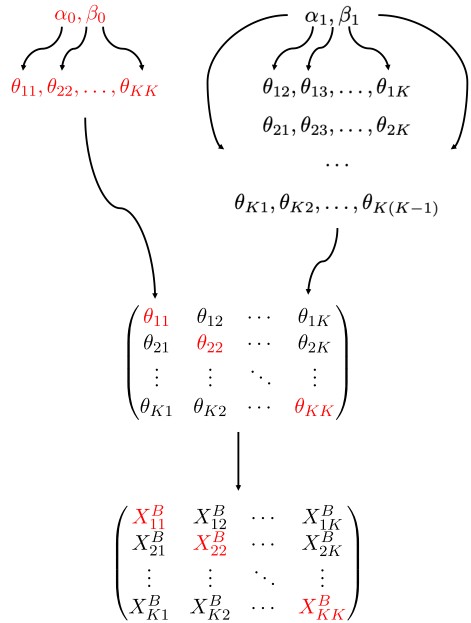

**Figure 1 A diagram of the hierarchical model.** The connectivity parameters $\theta_{ab}, a, b \in \{1, \ldots, K\}$, follow beta distributions of two sets of hyperparameters, *i.e.*, $(\alpha_0, \beta_0)$ for diagonal blocks (red) and $(\alpha_1, \beta_1)$ for off-diagonal blocks, and the number of edges $X_{ab}^B$ in a block, depends on $\theta_{ab}$ as in (3).

which leads to a high variance of the MLE. To stabilize the estimates, we may borrow information across blocks to improve estimation accuracy. To do this, we set up a hierarchical model by putting conjugate prior distributions on $\theta_{ab}$. To accommodate the heterogeneity in $\theta_{ab}$, we use two sets of hyperparameters so that the within and between-block connectivities are modeled separately:

$$\theta_{ab}|(\alpha_d, \beta_d) \sim \text{Beta}(\alpha_d, \beta_d), \quad a, b \in \{1, \ldots, K\}, \tag{5}$$

where $d = 0$ for $a = b$ and $d = 1$ for $a \neq b$, *i.e.*, the diagonal and off-diagonal elements of the connectivity matrix $\Theta$ follow $\text{Beta}(\alpha_0, \beta_0)$ and $\text{Beta}(\alpha_1, \beta_1)$, respectively. The prior distribution (5) together with (3) defines the distribution $[\mathbf{X}, \Theta|(\alpha_d, \beta_d)_{d=0,1}]$. Here $(\alpha_d, \beta_d)$, $d = 0, 1$, are hyperparameters to be estimated by our method. A diagram of our model is shown in Fig. 1. Note that the use of two sets of hyperparameters is in line with common assumptions of the stochastic block model, such as assortativity (*Danon et al., 2005*) or disassortativity, *i.e.*, within-group connectivities are different than between-group connectivities.

The conditional posterior distribution of $\theta_{ab}$ given $(X_{ab}^B, \alpha_d, \beta_d)$ is

$$\theta_{ab}|(X_{ab}^B, \alpha_d, \beta_d) \sim \text{Beta}(\alpha_d + X_{ab}^B, \beta_d + n_{ab} - X_{ab}^B),$$

and the conditional posterior mean of $\theta_{ab}$ is

$$\hat{\theta}_{ab}^{\text{EB}}(\alpha_d, \beta_d) \equiv \mathbb{E}(\theta_{ab}|X_{ab}^B, \alpha_d, \beta_d) \tag{6}$$

$$= \frac{\alpha_d + X_{ab}^B}{\alpha_d + \beta_d + n_{ab}} = \eta_{ab}\frac{\alpha_d}{\alpha_d + \beta_d} + (1 - \eta_{ab})\frac{X_{ab}^B}{n_{ab}},$$

for $a, b \in \{1, \ldots, K\}$, where

$$\eta_{ab} = \frac{\alpha_d + \beta_d}{\alpha_d + \beta_d + n_{ab}} \in [0, 1] \tag{7}$$

is the shrinkage factor that measures the amount of information borrowed across blocks. When the variance among $\theta_{ab}$ across the blocks is high, $\alpha_d$ and $\beta_d$ will be estimated to be small. Thus, $\eta_{ab}$ will be close to 0 so that the estimate $\hat{\theta}_{ab}^{\text{EB}}$ will be close to $\hat{\theta}_{ab}^{\text{MLE}}$. When the variance among $\theta_{ab}$ is low, our estimates of $\alpha_d$ and $\beta_d$ will be large, the shrinkage factor approaches 1, and eventually $\hat{\theta}_{ab}^{\text{EB}}$ will become identical across all blocks. In this case, we are essentially pooling data in all blocks to estimate $\theta_{ab}$. Generally speaking, the shrinkage factor $\eta_{ab}$ is determined by the data through the estimation of the hyperparameters $(\alpha_d, \beta_d)$, and it leads to a good compromise between the above two extreme cases.

Given the partition $Z$ from a graph clustering algorithm, we maximize the marginal likelihood of the observed adjacency matrix $\mathbf{X}$ to estimate the hyper-parameters $(\alpha_d, \beta_d)$ for $d = 0, 1$. Let $\mathbf{X}_{ab}$ denote the adjacency submatrix for nodes in the block $(a, b)$ defined by the partition $Z$. Integrating over $\Theta$, the marginal log-likelihood function for the diagonal blocks is

$$\mathscr{L}(\alpha_0, \beta_0|\mathbf{X}, Z) = \sum_{a=1}^{K} \log \mathbb{P}(\mathbf{X}_{aa}|\alpha_0, \beta_0)$$

$$= \sum_{a=1}^{K} \log \int_{\theta_{aa}} \mathbb{P}(\mathbf{X}_{aa}|\theta_{aa}) p(\theta_{aa}|\alpha_0, \beta_0) d\theta_{aa} \tag{8}$$

$$= \sum_{a=1}^{K} \log \text{Beta}(\alpha_0 + X_{aa}^B, \beta_0 + n_{aa} - X_{aa}^B) - K \log \text{Beta}(\alpha_0, \beta_0),$$

where $\text{Beta}(x, y) = \int_0^1 t^{x-1}(1-t)^{y-1}dt$ is the beta function. Similarly, the marginal log-likelihood function for the off-diagonal blocks is

$$\mathscr{L}(\alpha_1, \beta_1|\mathbf{X}, Z)$$

$$= \sum_{a<b} \log \text{Beta}(\alpha_1 + X_{ab}^B, \beta_1 + n_{ab} - X_{ab}^B) - \frac{1}{2}K(K-1)\log \text{Beta}(\alpha_1, \beta_1). \tag{9}$$

We find the maximum likelihood estimates of the hyper parameters, *i.e.*,

$$(\hat{\alpha}_d, \hat{\beta}_d) = \arg\max_{\alpha_d, \beta_d} \mathscr{L}(\alpha_d, \beta_d|\mathbf{X}, Z), \tag{10}$$

for $d = 0, 1$. Then we can estimate $\Theta$ by plugging the MLE of the hyper-parameters in (10) into (6), *i.e.*,

$$\hat{\theta}_{ab}^{\text{EB}} = \begin{cases} \hat{\theta}_{aa}^{\text{EB}}(\hat{\alpha}_0, \hat{\beta}_0), & a = b \\ \hat{\theta}_{ab}^{\text{EB}}(\hat{\alpha}_1, \hat{\beta}_1), & a \neq b \end{cases}.$$

(11)

Since the hyper-parameters are estimated using all blocks, our empirical Bayes estimates of $\theta_{ab}$ also make use of information from all data to improve the accuracy. Though (10) does not have a closed form solution, we can use an optimization algorithm such as bounded limited-memory BFGS (L-BFGS-B) (*Byrd et al., 1995*) to find the maximizer. The optimization algorithm starts at a random initial point, and we re-run the algorithm if it fails to converge. The log-likelihood functions in (8) and (9) are not necessarily concave, and thus finding the global maximizers cannot be guaranteed in theory. However, as shown in Fig. S2 in Supplemental Material, for a typical dataset the maximizers over a reasonable range of $(\alpha_d, \beta_d)_{d=0,1}$ can be easily found.

*Suwan et al. (2016)* developed a different empirical Bayesian method for SBMs under a random dot product graph formulation. They introduce $K$ latent positions, $v_1, \ldots, v_K \in \mathbb{R}^d$, and define the connection probabilities by inner products between the latent positions, $\theta_{ab} = \langle v_a, v_b \rangle$ for $1 \leq a, b \leq K$. The prior distribution for $v_k$ is a multivariate Gaussian distribution $v_k \sim \mathcal{N}_d(\hat{\mu}_k, \hat{\Sigma}_k)$. In particular, the parameters $\hat{\mu}_k, \hat{\Sigma}_k$ in the prior are chosen by Gaussian mixture modeling of pre-estimated latent positions obtained *via* adjacency spectral embedding. Thus, these prior distributions are called *empirical* priors and they are used to model the uncertainty in the latent positions $v_1, \ldots, v_K$. In our method, the hyperparameters $(\alpha, \beta)$ in the beta prior distributions are not pre-estimated by a separate method, but instead are estimated under a coherent hierarchical model. In addition to modeling uncertainty in the connectivity probabilities $\theta_{ab}$, the hyperparameters also lead to information sharing *via* shrinkage.

## Selecting partitions

So far we have regarded the number of blocks $K$ as given in our empirical Bayes method. The choice of $K$ will certainly impact the performance of our method. If $K$ is too small, for SBM many blocks will not be identified, and for graphon the approximated function will only have a small number of constant pieces, both leading to highly biased estimates. On the other hand, if $K$ is too big, the number of vertices in each block will be very small, resulting in high variances. Thus, it is important to select a proper number of blocks to achieve the best estimation accuracy.

Our empirical Bayes approach under the hierarchical model also provides a useful criterion for this model selection problem. Note that (8) and (9) define the conditional likelihood of $\mathbf{X}$ given the hyperparameters $(\alpha_d, \beta_d)$ and the partition $Z$ input from a graph clustering algorithm. We can compare this likelihood for different input partitions and select the best one.

Suppose we have $m$ candidate partition schemes $Z_1, \ldots, Z_m$. Denote the corresponding number of communities by $K_1, \ldots, K_m$. Our goal is to choose the optimal partition that maximizes the joint likelihood of the observed adjacency matrix $\mathbf{X}$ and the partition $Z$

with a penalty on the model complexity. To do this, we include $Z$ in our model as in (2) and put a Jeffreys prior (*Jeffreys, 1946*) on $\pi$, *i.e.*,

$$\pi \sim \text{Dirichlet}(\tau_1, \dots, \tau_K), \quad \tau_1 = \dots = \tau_K = 1/2.$$

The Jeffrey's prior is a standard non-informative prior that is invariant to re-parameterization. In general, $\tau_k = \tau$ for any $\tau \in (0, 1]$ is a common choice for a non-informative prior, with negligible effect on the posterior inference or model selection when the network size $n$ is large. Nonetheless, we could also use informative prior if strong prior knowledge is provided, for example, on $\pi$ or the expected community sizes.

For a partition $Z$ with $K$ communities, the joint likelihood of $\mathbf{X}$ and $Z$ given the hyper-parameters $(\alpha_0, \alpha_1, \beta_0, \beta_1)$ is

$$
\begin{aligned}
&\mathbb{P}(\mathbf{X}, Z | \alpha_0, \alpha_1, \beta_0, \beta_1) \\
&= \mathbb{P}(\mathbf{X} | Z, \alpha_0, \alpha_1, \beta_0, \beta_1) \int \mathbb{P}(Z | \pi) p(\pi) d\pi \\
&= \mathbb{P}(\mathbf{X} | Z, \alpha_0, \alpha_1, \beta_0, \beta_1) \frac{\Gamma\left(\sum\limits_{i=1}^{K} \tau_i\right) \prod\limits_{i=1}^{K} \Gamma(n_i + \tau_i)}{\Gamma\left(n + \sum\limits_{i=1}^{K} \tau_i\right) \prod\limits_{i=1}^{K} \Gamma(\tau_i)},
\end{aligned}
\tag{12}
$$

after marginalizing out the parameter $\pi$, where $n_i$ is the number of nodes in cluster $i$ defined by the partition $Z$. Maximizing over the hyperparameters leads to the MLE $(\hat{\alpha}_0, \hat{\alpha}_1, \hat{\beta}_0, \hat{\beta}_1)$ defined in (10). Evaluating the likelihood (12) at the estimated hyperparameters, we define the goodness-of-fit part for our model selection criterion as

$$
\begin{aligned}
J_Z &= \log \mathbb{P}(\mathbf{X}, Z | \hat{\alpha}_0, \hat{\alpha}_1, \hat{\beta}_0, \hat{\beta}_1) \\
&= \sum_{d \in \{0,1\}} \mathscr{L}(\hat{\alpha}_d, \hat{\beta}_d | \mathbf{X}, Z) + \log \frac{\Gamma\left(\sum\limits_{i=1}^{K} \tau_i\right) \prod\limits_{i=1}^{K} \Gamma(n_i + \tau_i)}{\Gamma\left(n + \sum\limits_{i=1}^{K} \tau_i\right) \prod\limits_{i=1}^{K} \Gamma(\tau_i)},
\end{aligned}
\tag{13}
$$

where $\mathscr{L}(\hat{\alpha}_d, \hat{\beta}_d | \mathbf{X}, Z)$ is as in (8) and (9) for $d = 0, 1$. Following the ICL-like (integrated complete likelihood) criterion in *Mariadassou, Robin & Vacher (2010)*, we add two penalty terms to control model complexity: The first term corresponds to a penalty on the number of parameters in $\pi$ and the second the number of parameters in $\Theta$. Therefore, our model selection criterion is to choose the partition

$$
\hat{Z} = \underset{Z \in \{Z_1, \dots, Z_m\}}{\arg\max} \left\{ J_Z - \frac{1}{2}\left[(K-1)\log n + \frac{K(K+1)}{2}\log\frac{n(n-1)}{2}\right] \right\},
\tag{14}
$$

where $K$ is the number of clusters defined by the partition $Z$. As we have mentioned in the introduction, there are quite a few graph clustering algorithms, and the performance of many of them is highly dependent on the input number of partitions. Our criterion for selecting the number of clusters applies to any method used for the node clustering step, and thus it protects our method from inferior input node clustering results. The ICL model

selection criterion (14) is an approximation to the marginal log-likelihood $\log \mathbb{P}(\mathbf{X}|K)$ (*Mariadassou, Robin & Vacher, 2010*). The joint likelihood (13) depends on the EB estimates of the hyperparameters, which is unique to our hierarchical model, while the VBEM criterion (*Latouche, Birmele & Ambroise, 2012*) uses a standard SBM likelihood without a hierarchical structure or estimation of priors. We can easily apply other penalty terms in various model selection criteria to our likelihood, and fully expect similar behavior in terms of selecting the number of clusters, since most of them approximate in the same way as the marginal likelihood or the Bayes factor.

## Graphon estimate

Now we assume that the true model is a graphon as in (1). We use an SBM with $K$ blocks as an approximation to the graphon, *i.e.*, we approximate $W(u, v)$ by a piecewise constant function: We divide the unit interval $[0, 1]$ into $K$ pieces based on $\pi$ so that the length of the $k$-th piece is $\pi_k$. Let the endpoints of these pieces be $c_k = \sum_{i=1}^{k} \pi_i$ for $k = 1, \cdots, K$ and put $c_0 \equiv 0$. Then the graphon function defined on $[0, 1] \times [0, 1]$ is approximated by a $K \times K$ blockwise constant function,

$$\widetilde{W}(u, v) = \theta_{ab} \qquad \text{if}(u, v) \in [c_{a-1}, c_a) \times [c_{b-1}, c_b).$$

To estimate a graphon $W$, we first run a clustering algorithm to estimate a partition $Z$ and then apply the empirical Bayes method to obtain $\hat{\theta}_{ab}^{EB}$. Let $n_k$ denote the size of the the $k$-th cluster of vertices. We calculate its proportion to estimate $\pi_k$ by $\hat{\pi}_k = n_k/n$ and compute the cumulative proportion $\hat{c}_k = \sum_{i=1}^{k} \hat{\pi}_i$ for $k = 1, \cdots, K$. Define a binning function,

$$\text{bin}(x) = 1 + \sum_{k=1}^{K} \mathbb{I}(c_k \le x), \tag{15}$$

and the graphon $W$ is then estimated by

$$\widehat{W}(x, y) = \hat{\theta}_{\text{bin}(x),\text{bin}(y)}^{EB}, \qquad x, y \in [0, 1). \tag{16}$$

As shown by *Bickel & Chen (2009)*, the graphon is not identifiable in the sense that any measure-preserving transformation on $[0, 1]$ will define an equivalent random graph. Following their method, imposing the constraint that

$$g(x) = \int_0^1 W(x, y) dy$$

is nondecreasing leads to identifiability. For SBM approximation, the corresponding constraint is that

$$g(l) = \sum_{k=1}^{K} \pi_k \theta_{lk} \tag{17}$$

is nondecreasing in $l$. This constraint can be satisfied by relabeling the $K$ clusters of nodes.

As for the SBM, selecting a proper number of clusters $K$ is important for the estimation of a graphon. We will apply the same model selection criterion (14) to choose the optimal partition $Z$ and the associated $K$ among a collection of partitions.

## RESULTS

### Simulated data

In this section we present numerical results on graphs simulated from stochastic block models and graphon functions. We compare our method with other existing methods in terms of estimating connection probabilities and model selection for choosing the number of clusters.

For stochastic block models, we compare our estimated connectivity matrix $\widehat{\Theta}_{\text{EB}}$ (11) to the maximum likelihood estimate $\widehat{\Theta}_{\text{MLE}}$ as in (4) and the variational Bayes inference $\widehat{\Theta}_{\text{VBEM}}$ from *Latouche, Birmele & Ambroise (2012)*. Variational Bayes inference provides a closed-form approximate posterior distribution for $(\pi, \Theta)$ by minimizing the KL divergence between an approximated and the underlying distributions of $[Z|\mathbf{X}]$. It constructs point estimates for the parameters based on EM iterations (Supplemental Material). We compute the mean squared error (MSE)

$$MSE = \frac{1}{n(n-1)} \sum_{i=1}^{n} \sum_{j \neq i} (\widehat{\Theta}'_{ij} - \Theta'_{ij})^2 \qquad (18)$$

of an estimated $n \times n$ connection probability matrix $\widehat{\Theta}'$. Here, $\Theta' = (\Theta'_{ij})_{n \times n}$ is the true connection probability matrix among the $n$ nodes, i.e., $\Theta'_{ij} = \theta_{ab}$ if $Z^*(i) = a$ and $Z^*(j) = b$ for $i, j = 1, \ldots, n$, where $Z^*$ is the true partition, and $\widehat{\Theta}'_{ij} = \hat{\theta}_{ab}$ if $Z(i) = a$ and $Z(j) = b$. For graphons, $\widehat{W}(x, y)$ is estimated by SBM approximation, and correspondingly the mean integrated squared error is calculated as

$$MSE = \int_0^1 \int_0^1 (W(x, y) - \widehat{W}(x, y))^2 dxdy. \qquad (19)$$

Due to the nonidentifiability of graphons, the MSE is calculated after relabeling node clusters based on the constraint (17) to make $\widehat{W}$ comparable to $W$.

We compare our model selection criterion (14) to the variational Bayes method developed by *Latouche, Birmele & Ambroise (2012)* (VBEM) and the cross validation risk of precision parameter (CVRP) in *Airoldi, Costa & Chan (2013)*. The CVRP is defined as

$$\mathscr{I}_{\text{CVRP}}(K) = \frac{2K}{n-1} - \frac{(n+1)K}{n-1} \sum_{i=1}^{K} \left( \frac{n_i}{n} \right)^2, \qquad (20)$$

where $n_i$ is the number of vertices in group $i$. Then, the number of clusters $K$ is selected by minimizing the risk $\mathscr{I}_{\text{CVRP}}$, i.e.,

$$\hat{K}_{\text{CVRP}} = \arg\min_{K} \mathcal{J} CVRP(K). \tag{21}$$

We use $\mathcal{J}_{EB}$, $\mathcal{J}_{VBEM}$ and $\mathcal{J}_{CVRP}$ to denote, respectively, the three criteria mentioned above.

### Results on SBM: homogeneous block connectivity

We designed a constrained SBM that generates affiliation networks, *i.e.*, two vertices within the same community connect with probability $\lambda$, and from different communities with probability $\varepsilon < \lambda$. We also added a parameter $\rho \in (0,1]$ to control the sparsity of the graph. The corresponding true connectivity matrix is

$$\Theta^* = \rho \begin{pmatrix} \lambda & \varepsilon & \cdots & \varepsilon \\ \varepsilon & \lambda & \cdots & \vdots \\ \vdots & & \ddots & \varepsilon \\ \varepsilon & \cdots & \varepsilon & \lambda \end{pmatrix}_{K^* \times K^*},$$

where $K^*$ is the number of communities.

To generate dense graphs (model 1), we set $\lambda = 0.9$, $\varepsilon = 0.1$, and $\rho = 1$. We generated graphs with $n = 200$ vertices and the number of communities $K^* \in \{10, 11, \ldots, 18\}$. For each choice of $K^*$, we generated 100 networks independently. For each network, all the nodes were randomly divided into $K^*$ clusters with equal probability $1/K^*$, and then connected according to the connectivity matrix $\Theta^*$ and their cluster labels. Note that the simulated node clusters had very different sizes, ranging between 7 and 35, due to the high variance in block size.

We also used $\lambda = 0.9$, $\varepsilon = 0.1$ and $\rho = 0.2$ to generate sparse graphs (model 2), while keeping $K^* = 10$ but changing the network size $n \in \{200, 250, 300, 350, 400, 450\}$. For each network size $n$, we followed the same procedure as in model 1 and generated 100 networks independently. Here "sparse" refers to a lower edge density around 0.035, which is 20% of the graphs generated in model 1.

For a simulated graph, we applied the variational Bayes algorithm (*Latouche, Birmele & Ambroise, 2012*) with an input number of clusters $K = 1, \ldots, 20$, from which we obtained $K$ communities and a Bayesian estimate $\widehat{\Theta}_{VBEM}(K)$ of the connecting probabilities among the $K \times K$ blocks. Given the estimated communities by the variational Bayes algorithm, we found $\widehat{\Theta}_{MLE}(K)$ as in (4) and our empirical Bayes estimate $\widehat{\Theta}_{EB}(K)$ as in (11) and compared them to the VBEM estimate. As the estimates are functions of $K$, so are their MSEs as defined in (18). Let $MSE_{MLE}(K)$ be the mean squared error of the MLE by plugging $\widehat{\Theta}_{MLE}(K)$ into (18), where each element $\widehat{\Theta}'_{ij}$ is given by $\widehat{\Theta}_{MLE}(K)$ and the partition $Z$. Then we define $\tilde{K}$ as the number of clusters that minimizes the MSE of the MLE, *i.e.*,

$$\tilde{K} = \arg\min_{K} MSE_{MLE}(K) \tag{22}$$

over the input range of $K$. For the 100 graphs generated under the same matrix $\Theta^*$, they share the same $K^*$ while each one of them defines a corresponding $\tilde{K}$. Both $K^*$ and $\tilde{K}$ were used in our comparisons on model selection criteria for the number of blocks. In

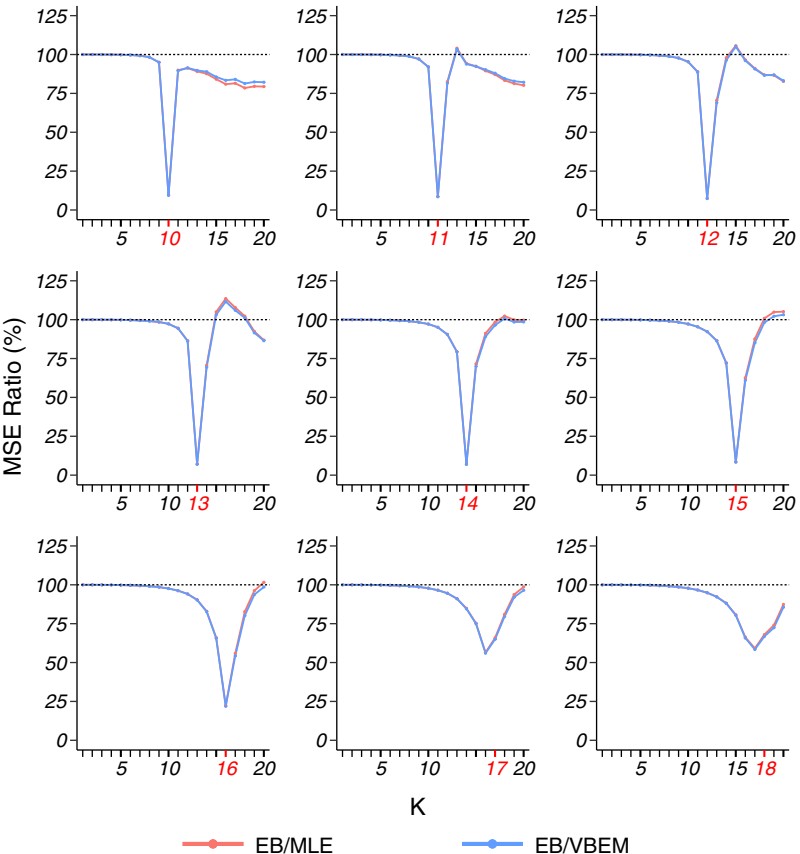

**Figure 2 MSE ratios in model 1 simulation.** The true number of blocks $K^*$ (marked in red) ranges from 10 to 18 and the results for graphs with each $K^*$ are shown in a panel. For the 100 graphs generated under each $K^*$, the MSE ratios of the estimates $\widehat{\Theta}_{\text{MLE}}$ and $\widehat{\Theta}_{\text{VBEM}}$ over $\widehat{\Theta}_{\text{EB}}$ are plotted against the input number of blocks $K$ chosen in the clustering step.

particular, for a general graphon, $K^*$ may not be clearly defined and in such a case, $\tilde{K}$ serves as the reference for comparison.

For dense graphs (model 1), as shown in Fig. 2, we compared the MSEs (18) of the three estimates of $\Theta$ to the true connectivity matrix and presented the ratio of the MSE of our EB estimate to the MSEs of the MLE and VBEM estimate. For dense stochastic block models, the accuracy of MLE and that of VBEM were close, whereas EB gave better estimates for almost all $K$ values, *i.e.*, MSE ratios were smaller than 100%. We see a significantly smaller MSE ratio when $K$ is close to $K^*$, especially when $K^*$ is relatively small. For example, the MSE ratios EB/MLE and EB/VBEM were lower than 10% at $K = K^*$ when $K^* = 10, \ldots, 15$. When $K^*$ went bigger, such as $K^* = 17, 18$ in the simulation, the $\tilde{K}$ for most of the graphs was less than $K^*$, and the MSE ratios reached a minimum level at some $K < K^*$, which was slightly above 50%.

Table 1 presents the model selection results on the simulated dense graphs from model 1, where we define $E_{K^*}$ and $E_{\tilde{K}}$ as the average deviation of the selected number of blocks $\hat{K}$ from $K^*$ and from $\tilde{K}$ respectively, *i.e.*,

**Table 1** Model selection comparison for model 1 among the $\hat{K}$ chosen by (a) CVRP, (b) VEBM, and (c) EB.

| $K^*\backslash\hat{K}$ | 8 | 9 | 10 | 11 | 12 | 13 | 14 | 15 | 16 | 17 | 18 | $E_{K^*}$ | $E_{\tilde{K}}$ |
|---|---|---|---|---|---|---|---|---|---|---|---|---|---|
| **(a) CVRP** | | | | | | | | | | | | | |
| 10 | | 99 | 1 | | | | | | | | | 0.99 | 0.99 |
| 11 | | | 100 | | | | | | | | | 1.00 | 1.00 |
| 12 | | | 3 | 96 | 1 | | | | | | | 1.02 | 1.02 |
| 13 | | | | | 67 | 33 | | | | | | 0.67 | 0.67 |
| 14 | | | | | 6 | 93 | 1 | | | | | 1.06 | 1.06 |
| 15 | | | | | | 23 | 77 | | | | | 1.23 | 1.26 |
| 16 | | | | | | 2 | 13 | 85 | | | | 1.17 | 1.31 |
| 17 | | | | | | | 1 | 29 | 70 | | | **1.31** | **1.33** |
| 18 | | | | | | | | 3 | 87 | 10 | | **1.93** | **1.27** |
| **(b) VBEM** | | | | | | | | | | | | | |
| 10 | | | 100 | | | | | | | | | **0.00** | **0.00** |
| 11 | | | | 100 | | | | | | | | **0.00** | **0.00** |
| 12 | | | | | 100 | | | | | | | **0.00** | **0.00** |
| 13 | | | | | | 100 | | | | | | **0.00** | **0.00** |
| 14 | | | | | | 4 | 96 | | | | | 0.04 | 0.45 |
| 15 | | | | | 1 | 2 | 35 | 62 | | | | 0.39 | 0.85 |
| 16 | | | | | | 1 | 28 | 53 | 18 | | | 1.12 | 1.26 |
| 17 | | | | | | | 6 | 53 | 35 | 6 | | 2.59 | 2.61 |
| 18 | | | | | 1 | | 7 | 32 | 44 | 16 | | 3.33 | 2.67 |
| **(c) EB** | | | | | | | | | | | | | |
| 10 | | | 100 | | | | | | | | | **0.00** | **0.00** |
| 11 | | | | 100 | | | | | | | | **0.00** | **0.00** |
| 12 | | | | | 100 | | | | | | | **0.00** | **0.00** |
| 13 | | | | | | 100 | | | | | | **0.00** | **0.00** |
| 14 | | | | | | | 100 | | | | | **0.00** | **0.00** |
| 15 | | | | | | | 1 | 99 | | | | **0.01** | **0.04** |
| 16 | | | | | | | | 30 | 70 | | | **0.30** | **0.44** |
| 17 | | | | | | | | 33 | 67 | | | 1.33 | 1.35 |
| 18 | | | | | | | | 1 | 95 | 4 | | 1.97 | 1.31 |

**Note:**
Each row in a table reports the frequency of $\hat{K}$ across 100 graphs. The last two columns report two mean absolute deviations, the minimum of which among the three methods is in bold for each $K^*$.

$$E_{K^*} = \frac{1}{M}\sum_{t=1}^{M}|\hat{K}_t - K^*|, \qquad E_{\tilde{K}} = \frac{1}{M}\sum_{t=1}^{M}|\hat{K}_t - \tilde{K}_t|, \tag{23}$$

where $t \in \{1,\dots,M\}$ is the index of the graphs generated under the same $\Theta^*$, $\hat{K}_t$ is the estimated number of clusters by a model selection criterion, and $\tilde{K}_t$ is the $\tilde{K}$ defined by (22) for the $t$-th graph. When $K^*$ was small, such as $10 \le K^* \le 13$, $\mathscr{I}_{VBEM}$ and $\mathscr{I}_{EB}$ gave the same results, and both accurately selected $\hat{K} = K^*$ as the optimal number of blocks. As $K^*$ increased, $\mathscr{I}_{EB}$ outperformed $\mathscr{I}_{VBEM}$, and was comparable to $\mathscr{I}_{CVRP}$ in terms of $E_{K^*}$. In fact, for a limited graph size $n = 200$ here, the average number of vertices in each block will be smaller as $K^*$ increases, making it hard for small communities to be detected.

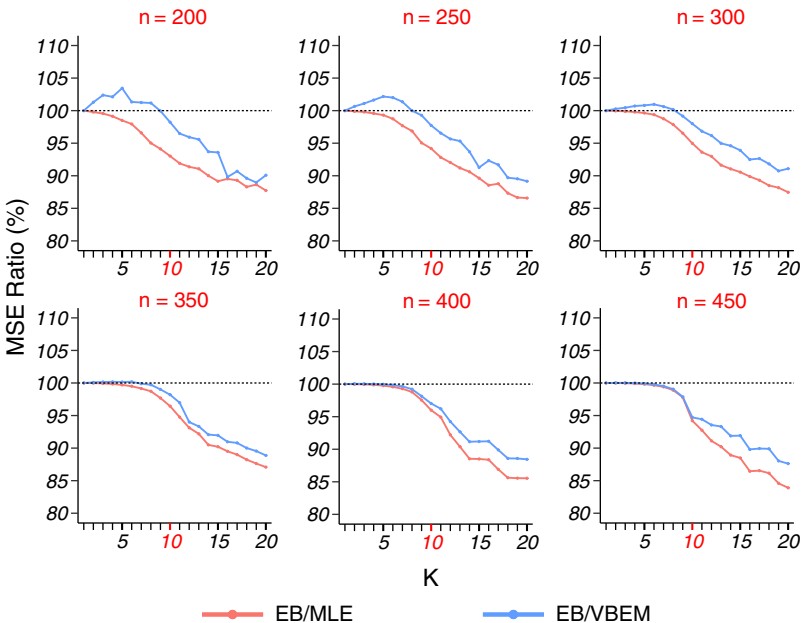

**Figure 3  MSE ratios in model 2 simulation.** The results for graphs with each network size $n$ are shown in a panel, plotted in the same format as Fig. 2.

Therefore, $\tilde{K}$ may better reflect the number of clusters that fit well the observed network. Considering this, we see $\mathscr{I}_{EB}$ had both smaller $E_{K^*}$ and $E_{\tilde{K}}$ than $\mathscr{I}_{VBEM}$ in general, which indicates the superiority of our model selection method. $\mathscr{I}_{CVRP}$ showed relatively stable performance in terms of $E_{K^*}$ and $E_{\tilde{K}}$, but the results were not satisfactory for small $K^*$. In summary, from the simulation results on dense graphs (model 1), EB has demonstrated the highest estimation accuracy, especially when the clustering algorithm finds the true number of communities, and the EB model selection criterion generally selects the best model.

Detecting the true number of blocks for a sparse graph (model 2) is harder because of fewer edge connections in a block. Thus, we fixed $K^* = 10$ and varied the network size $n$ from 200 to 450. In terms of estimation accuracy, Fig. 3 shows that our EB estimate had better performance than MLE in almost all the cases (except when $K = 1$ under which the two estimates were identical), and the MSE ratio kept decreasing as $K$ increased. In particular, for $K = K^* = 10$, the MSE ratio of EB over MLE was about 95%. If the number of blocks is overestimated (say $K > 15$), the MSE ratio can drop to <90%. When compared to VBEM, for a small network size $n$ and a small number of blocks $K$, EB estimates can be slightly less accurate (<5% increase in MSE), but as $K$ increases and becomes close to $K^*$, the MSE ratio decreases to the same level as that of EB over MLE. As reported in Table 2, for all the cases $\mathscr{I}_{EB}$ achieved the best model selection performance with the smallest $E_{K^*}$ and $E_{\tilde{K}}$ among the three methods. This highlights the usefulness of our model selection criterion for the more challenging sparse graph settings.

More detailed results for both models 1 and 2 in this simulation study can be found in the Supplemental Material.

**Table 2 Model selection comparison for model 2 ($K^* = 10$) among the $\hat{K}$ chosen by (a) CVRP, (b) VEBM, and (c) EB, in similar format as Table 1.**

| $n\backslash\hat{K}$ | 1 | 2 | 3 | 4 | 5 | 6 | 7 | 8 | 9 | 10 | 11 | 12 | $E_{K^*}$ | $E_{\tilde{K}}$ |
|---|---|---|---|---|---|---|---|---|---|---|---|---|---|---|
| **(a) CVRP** | | | | | | | | | | | | | | |
| 200 | 100 | | | | | | | | | | | | 9 | 2.84 |
| 250 | 100 | | | | | | | | | | | | 9 | 6.86 |
| 300 | 95 | | | | | | | | 1 | 4 | | | 8.56 | 8.84 |
| 350 | 71 | | | | | | | 1 | 14 | 14 | | | 6.55 | 8.17 |
| 400 | 37 | | | | | | | | 28 | 35 | | | 3.61 | 5.21 |
| 450 | 17 | | | | | | | | 11 | 71 | 1 | | 1.65 | 2.50 |
| **(b) VBEM** | | | | | | | | | | | | | | |
| 200 | 28 | 51 | 19 | 2 | | | | | | | | | 8.05 | 2.18 |
| 250 | | 8 | 30 | 42 | 13 | 6 | | | 1 | | | | 6.16 | 4.04 |
| 300 | | 1 | 11 | 31 | 37 | 20 | | | | | | | 4.36 | 4.59 |
| 350 | | | | | 14 | 43 | 36 | 7 | | | | | 2.64 | 4.22 |
| 400 | | | | | | 3 | 34 | 47 | 14 | 1 | 1 | | 1.27 | 2.83 |
| 450 | | | | | | 1 | 3 | 37 | 52 | 6 | 1 | | 0.54 | 1.25 |
| **(c) EB** | | | | | | | | | | | | | | |
| 200 | | 6 | 12 | 24 | 29 | 24 | 4 | 1 | | | | | **5.31** | **2.09** |
| 250 | | | 6 | 21 | 38 | 21 | 12 | 2 | | | | | **3.82** | **2.20** |
| 300 | | | | 1 | 13 | 32 | 35 | 18 | 1 | | | | **2.41** | **2.74** |
| 350 | | | | | | 2 | 31 | 47 | 20 | | | | **1.15** | **2.81** |
| 400 | | | | | | | | 10 | 38 | 48 | 3 | 1 | **0.63** | **2.13** |
| 450 | | | | | | | | 2 | 13 | 78 | 7 | | **0.24** | **0.97** |

**Note:**
Each row in a table reports the frequency of $\hat{K}$ across 100 graphs. The last two columns report two mean absolute deviations, the minimum of which among the three methods is in bold for each $K^*$.

### Results on SBM: heterogeneous block connectivity

In this section, we show how the performance changes when heterogeneous block connectivity probabilities are used. We consider the following connectivity matrix

$$
\Theta^* = \rho \begin{pmatrix} \lambda_1 & \varepsilon_{12} & \cdots & & \varepsilon_{1K^*} \\ \varepsilon_{21} & \lambda_2 & \cdots & & \vdots \\ \vdots & & \ddots & & \varepsilon_{(K^*-1)K^*} \\ \varepsilon_{K^*1} & \cdots & \varepsilon_{K^*(K^*-1)} & & \lambda_{K^*} \end{pmatrix}_{K^* \times K^*},
$$

where the values are sampled from uniform distributions.

Similar to model 1, to generate dense graphs (model 1s), we set $\rho = 1$, and drew $\lambda_i \sim U(0.5, 0.9)$ for $i \in \{1, \ldots, K^*\}$ and $\varepsilon_{ij} \sim U(0.3, 0.5)$ for $i, j \in \{1, \ldots, K^*\}$, $i \neq j$. We generated graphs with $n = 200$ vertices and the number of communities $K^* \in \{10, 11, \ldots, 18\}$. For each choice of $K^*$, we generated 100 networks independently with parameters sampled from the above uniform distributions. For each network, the nodes were randomly divided into $K^*$ clusters with equal probability $1/K^*$, and then connected according to the

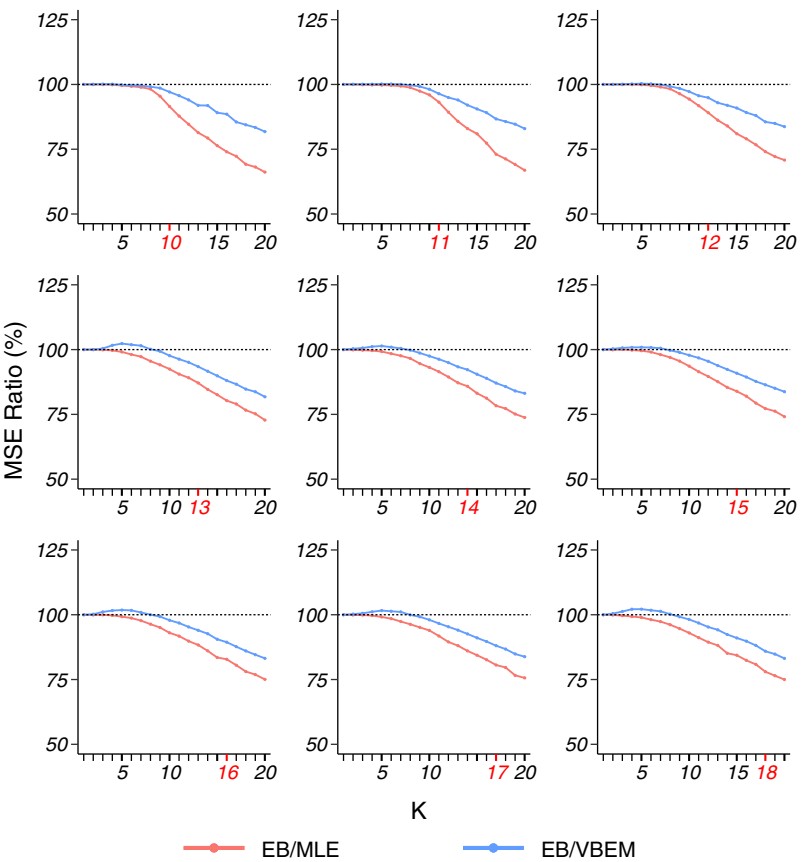

**Figure 4** MSE ratios in model 1s simulation, plotted in the same format as **Fig. 2.**

connectivity matrix $\Theta^*$ and their cluster labels. Overall, EB outperformed MLE and VBEM with respect to MSE. Different from model 1 where the smallest ratios of EB/MLE and EB/VBEM were observed at $K = K^*$ for most of the simulations, the MSE ratio of EB over MLE and VBEM decreases smoothly with the increase in $K$ (Fig. 4). In terms of model selection, EB was better than VBEM when $K^* \geq 14$ and comparable to VBEM with smaller $K$, although the improvement was slightly less substantial. The detailed results are reported in Table 3.

We also used $\rho = 0.2$ and the same setting for $\lambda$ and $\varepsilon$ as above to generate sparse graphs (model 2s), while keeping $K^* = 10$ but changing the network size $n \in \{200, 250, 300, 350, 400, 450\}$. For each network size $n$, we followed the same procedure as in model 1s and generated 100 networks independently. The results are similar to the homogeneous case (model 2). The detailed results are provided in the Supplemental Material.

### Results on graphon model

Following the same design as in *Latouche & Robin (2016)*, we choose a graphon function

$$W(x, y) = \rho \lambda^2 (xy)^{\lambda - 1}$$

with two parameters $\lambda \leq 1/\sqrt{\rho}$. Here, $\rho$ controls the sparsity of the graph, as the

**Table 3** Model selection comparison for model 1s among the $\hat{K}$ chosen by (a) CVRP, (b) VEBM, and (c) EB, in the same format as Table 1.

| $K^*\backslash\hat{K}$ | 1 | 2 | 3 | 4 | 5 | 6 | 7 | 8 | 9 | 10 | 11 | 12 | 13 | 14 | 15 | 16 | 17 | 18 | 19 | 20 | $E_{K^*}$ | $E_{\tilde{K}}$ |
|---|---|---|---|---|---|---|---|---|---|---|---|---|---|---|---|---|---|---|---|---|---|---|
| (a) CVRP | | | | | | | | | | | | | | | | | | | | | | |
| 10 | 2 | | | | | | 1 | 7 | 42 | 46 | 2 | | | | | | | | | | 0.79 | 0.88 |
| 11 | 3 | | | | | | 1 | 10 | 44 | 42 | | | | | | | | | | | 0.97 | 1.15 |
| 12 | 1 | | | | | | | 2 | 3 | 15 | 48 | 29 | 1 | | 1 | | | | | | 1.1 | 1.55 |
| 13 | 14 | | | | | | 1 | 2 | 3 | 15 | 22 | 29 | 12 | 1 | 1 | | | | | | 3.17 | 3.51 |
| 14 | 23 | | | | | | | | 1 | 16 | 18 | 22 | 15 | 5 | | | | | | | 4.81 | 5.01 |
| 15 | 41 | | | | | | | | 1 | 3 | 7 | 18 | 11 | 14 | 4 | 1 | | | | | 7.14 | 6.94 |
| 16 | 45 | 1 | | | | | | 1 | | 3 | 10 | 15 | 13 | 6 | 5 | | 1 | | | | 8.82 | 7.61 |
| 17 | 53 | | 1 | | | | | 1 | | 1 | 5 | 9 | 12 | 7 | 4 | 3 | 2 | 2 | | | 10.34 | 8.53 |
| 18 | 80 | 1 | | 1 | | | | | | 1 | 1 | 4 | 2 | 7 | 1 | | 2 | | | | 14.71 | 10.37 |
| (b) VBEM | | | | | | | | | | | | | | | | | | | | | | |
| 10 | | | | | | | 1 | 5 | 88 | 6 | | | | | | | | | | | **0.13** | **0.1** |
| 11 | | | | | | | | 1 | 15 | 81 | 3 | | | | | | | | | | **0.2** | **0.32** |
| 12 | | | | | | | | 1 | 8 | 27 | 56 | 8 | | | | | | | | | **0.54** | **0.85** |
| 13 | | | | | | | | 2 | 6 | 15 | 30 | 25 | 18 | 3 | | 1 | | | | | **1.7** | **2.02** |
| 14 | | | | | | | | 1 | 8 | 27 | 13 | 33 | 16 | 2 | | | | | | | 2.75 | 2.93 |
| 15 | | | | | | | 1 | 1 | 9 | 17 | 34 | 19 | 13 | 5 | | 1 | | | | | 3.79 | 3.43 |
| 16 | | | | | | | | 10 | 13 | 18 | 27 | 18 | 9 | 4 | | 1 | | | | | 5.21 | 3.94 |
| 17 | | | | | 1 | | 2 | 6 | 19 | 17 | 28 | 9 | 10 | 5 | 2 | 1 | | | | | 6.3 | 4.27 |
| 18 | | | 1 | | 3 | 4 | 12 | 24 | 16 | 15 | 11 | 8 | 4 | 1 | 1 | | | | | | 8.9 | 4.38 |
| (c) EB | | | | | | | | | | | | | | | | | | | | | | |
| 10 | | | | | | | | | 2 | 81 | 13 | 1 | | | | | | | | | 0.17 | 0.16 |
| 11 | | | | | | | | | | 8 | 76 | 15 | 1 | | | | | | | | 0.25 | 0.36 |
| 12 | | | | | | | | 1 | 5 | 5 | 8 | 66 | 13 | 4 | | | | | | | 0.58 | 0.89 |
| 13 | | | | | | | | 1 | | 6 | 8 | 10 | 16 | 19 | 15 | 23 | 1 | 1 | | | 1.76 | 2.27 |
| 14 | | | | | | | 1 | 2 | 3 | 1 | 2 | 11 | 18 | 17 | 17 | 23 | 3 | 1 | 1 | | **1.63** | **2.19** |
| 15 | | | | | | | 2 | 3 | 3 | 5 | 8 | 8 | 12 | 19 | 25 | 14 | | 1 | | | **1.96** | **2.48** |
| 16 | | | | | 1 | | | 2 | 2 | 4 | 10 | 9 | 10 | 18 | 24 | 15 | 3 | 2 | | | **2.48** | **2.52** |
| 17 | | | | | | 1 | 1 | 3 | 2 | 1 | 10 | 11 | 12 | 19 | 22 | 16 | 1 | 1 | | | **3.56** | **2.4** |
| 18 | | | | | | | 1 | 3 | 5 | 10 | 9 | 5 | 17 | 14 | 17 | 18 | 1 | | | | **4.88** | **2.73** |

**Note:**
Each row in a table reports the frequency of $\hat{K}$ across 100 graphs. The last two columns report two mean absolute deviations, the minimum of which among the three methods is in bold for each $K^*$.

expected number of edges is proportional to $\rho$, and $\lambda$ controls the concentration of the degrees, so that more edges will concentrate on fewer nodes if $\lambda$ is large. We chose $\rho \in \{10^{-1}, 10^{-1.5}, 10^{-2}\}$ and $\lambda \in \{2, 3, 5\}$, and simulated graphs of size $n = 100$ (model 3) and of size $n = 316$ ($\approx 10^{2.5}$) (model 4). For each network, we used SBM approximation with the number of clusters $K = 1, 2, \ldots, 10$. Using (22), we also defined $\tilde{K}$ as the number of blocks that minimizes the mean squared error (19) of the MLE, i.e., $\tilde{K}$ is the number of communities that best fits the observed network. Figure 5 shows the graphon function for some values of $(\rho, \lambda)$. The parameter $\rho$ controls the scale of the function, and thus the grophon functions reach the maximum height when $\rho = 10^{-1}$. While not shown in the

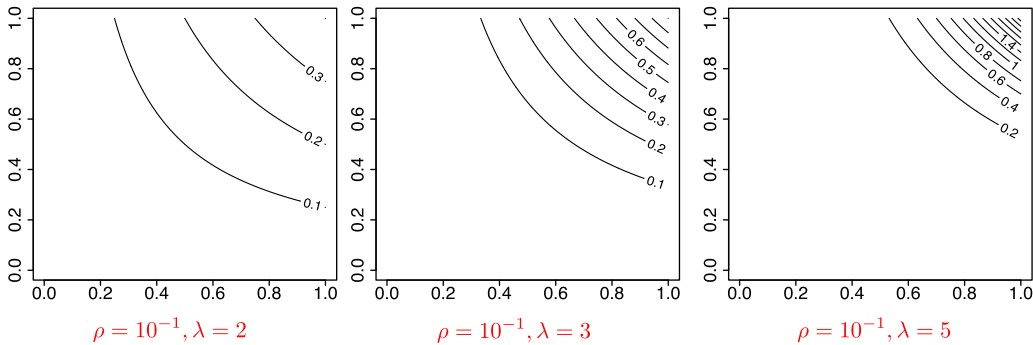

**Figure 5** Visualization of the graphon function $W(x, y) = \rho \lambda^2 (xy)^{\lambda-1}$ in model 3 and 4.

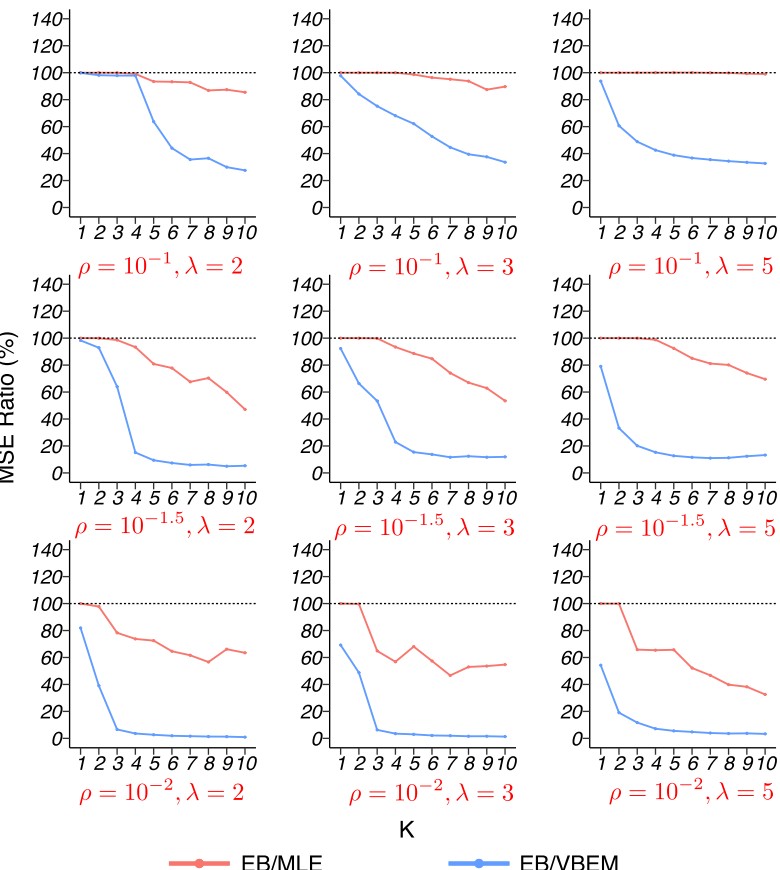

**Figure 6** MSE ratios in model 4 simulation with graph size $n = 316$. The results for graphs with each combination of $\rho$ and $\lambda$ are shown in a panel.

figure, for $\rho = 10^{-1.5}$ or $10^{-2}$ the functions are scaled down and have lower values. Meanwhile, $\lambda$ controls the concentration of the function, such that a graphon defined by a higher value of $\lambda$ shows a highly concentrated peak as for $\lambda = 5$ in the figure.

The MSE ratios between our EB estimate and the other two competing methods, MLE and VBEM, are shown in Fig. 6 for graphs of size $n = 316$. The results for $n = 100$ are similar and relegated to the Supplemental Material. In general, our EB method achieved

**Table 4 Model selection comparison for graphons.** (Reported is the mean absolute deviation $E_{\tilde{K}}$ for graphs generated under each combination of $(\rho, \lambda)$.

| | | n = 100 | | | n = 316 | | |
|---|---|---|---|---|---|---|---|
| | | CVRP | VBEM | EB | CVRP | VBEM | EB |
| $\rho = 10^{-1}$ | $\lambda = 2$ | 1.16 | **0.96** | 1.11 | 4.92 | 2.55 | **2.38** |
| | $\lambda = 3$ | 5.42 | **1.54** | 2.03 | 5.8 | 1.92 | **1.91** |
| | $\lambda = 5$ | 3.88 | **1.28** | 1.63 | 7.43 | 1.66 | **1.50** |
| $\rho = 10^{-1.5}$ | $\lambda = 2$ | 2.01 | 1.86 | **1.83** | 4.76 | 3.72 | **3.70** |
| | $\lambda = 3$ | 1.81 | 1.02 | **0.95** | 3.93 | 2.02 | **1.96** |
| | $\lambda = 5$ | 2.05 | 1.03 | **0.98** | 4.58 | **1.60** | 1.79 |
| $\rho = 10^{-2}$ | $\lambda = 2$ | 0.86 | **0.85** | 0.86 | 2.56 | **2.24** | 2.25 |
| | $\lambda = 3$ | **1.41** | 1.45 | 1.48 | 1.48 | 1.35 | **1.31** |
| | $\lambda = 5$ | **1.52** | 1.61 | 1.7 | 2.77 | 1.72 | **1.67** |

**Note:**
The minimal $E_{\tilde{K}}$ among the three methods is highlighted in bold.

higher accuracy with smaller MSEs than the other two methods. For most cases, our EB estimate was more accurate than the MLE, with the MSE ratios between 60% and 100%. Compared to VBEM, our EB estimate achieved substantially smaller MSEs with ratios below 20%. For both graph sizes, the improvement of the EB method over the other two competitors was especially significant when the graph was sparse ($\rho$ small). In such a case, fewer connections between nodes are observed in a network, and there is a high probability to have zero edge within the cluster. For blocks with lower connectivity, MLE tends to underestimate their connectivity, while shrinkage helps the situation by borrowing information from other blocks.

The model selection results are reported in Table 4. Since the true number of communities under the graphon model is not clearly defined, we used $\tilde{K}$ as the ground-truth to evaluate model selection performance. For both $n = 100$ and $n = 316$, the mean absolute deviation $E_{\tilde{K}}$ (23) of the $\hat{K}$ selected by our criterion $\mathscr{I}_{EB}$ was either the smallest or was very close to the smallest value among the three methods. While EB and VBEM were generally comparable, CVRP showed unstable performance as its $E_{\tilde{K}}$ could be much larger than the other two methods in some cases (such as $\rho = 10^{-1}$ and $\rho = 10^{-1.5}$). See Supplemental Material for more detailed results.

To expand the scope of this study, we further compared the performance of our EB method on graphons with a non-Bayesian approach. A commonly used algorithm is network histogram approximation (NHA) developed by *Olhede & Wolfe (2014)*. The authors showed the universality of graphon approximation through regular stochastic block model and introduced an automatic bandwidth selection rule to select the best block model to represent graphon functions. The method fist divides degree-sorted vertices into equal-sized groups and selects the histogram bandwidth that maximizes the likelihood under an SBM. Given the automatically selected histogram bandwidth, the model parameters are estimated by the MLE in (4). In the comparison, we substitute the MLE estimate with our EB estimate to see if it can improve the accuracy.

**Table 5 Comparison of MSE of the graphon estimates by network histogram approximation (NHA) and empirical Bayes (EB).**

| | $n = 100$ | | | $n = 316$ | | |
|---|---|---|---|---|---|---|
| | NHA | EB | Improve % | NHA | EB | Improve % |
| $\rho = 10^{-1}, \lambda = 2$ | 0.00459 | 0.00351 | 23.5 | 0.00284 | 0.00230 | 19.1 |
| $\rho = 10^{-1}, \lambda = 3$ | 0.00223 | 0.00214 | 3.88 | 0.000671 | 0.000654 | 2.58 |
| $\rho = 10^{-1}, \lambda = 5$ | 0.0116 | 0.0116 | 0 | 0.00760 | 0.00756 | 0.55 |

In our simulation we used NHA to estimation the graphon functions in model 3 and 4 with $\rho = 10^{-1}$, using the suggested parameter ($c = 4$ in *Olhede & Wolfe (2014)*) to select the NHA bandwidth. NHA did not work on the sparser cases since too many nodes have a degree of zero. Table 5 shows that EB indeed improved the graphon estimation by NHA as well. The MSE for each set of parameters shown in the table is the average results from 100 networks. For $\lambda = 2$ in which case the graphon function has lower variability, EB outperformed MLE substantially. For larger $\lambda$'s, the two methods had comparable accuracy.

We briefly summarize a few key observations from the simulation studies. It is seen that EB estimates had smaller MSEs than the other two methods in most of the cases above. For the dense SBM (model 1), the accuracy of EB estimate was much higher. The relative low variance in connectivity across different blocks led to higher degree of shrinkage and information sharing among the EB estimates. For the sparse SBM (model 2), heterogenous SBM (model 1s and 2s) and graphon models (model 3 and 4), EB showed moderate improvements over the two competing methods in general. When the graph is sparse, EB can be much more accurate than VBEM, as shown in Fig. 6. As for model selection, EB generally selected the number of clusters $\hat{K}$ that was closer to $K^*$ and $\tilde{K}$ in all the models above, which demonstrates the usefulness of our hierarchical model for deriving likelihood-based model selection criterion.

*Alternative clustering and running time comparison*

Our results and numerical comparisons were conducted to demonstrate the uniform accuracy improvement: By varying the input number of clusters so some cluster results could be very inaccurate, our EB estimates reached smaller MSEs for almost all the clustering results. To further demonstrate this point, we also applied our EB estimates after spectral clustering. As shown in Fig. 7, our method improved the parameter estimation accuracy as well: Under the same simulation setting as in Figs. 2 and 3, the EB/MLE MSE ratio shows a similar pattern to the results of the previous simulation in SBMs.

The computation of our EB method is only the maximization of the likelihood (8, 9). The objective is the sum of two separate functions. Thus, we just need to maximize two bi-variate functions, regardless of the problem size ($n$, $K$). In general, the computation time is negligible compared to the graph clustering step. Table 6 reports the average running times (in seconds) of spectral clustering ($T_C$) and our EB estimation ($T_E$) by BFGS for various network size $n$ and number of communities $K$, on a single 2.6 GHz Intel i7 core. It

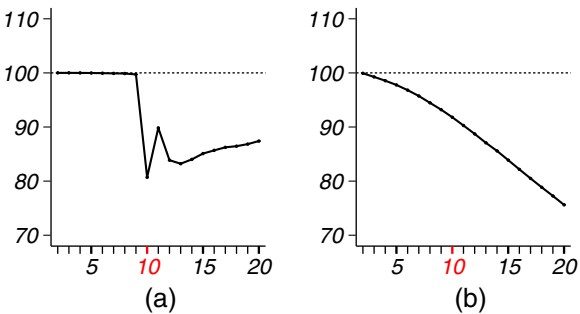

**Figure 7 MSE ratios of EB/MLE in spectral clustering simulation.** (A) model 1 with parameters $K^* = 10$, $n = 200$. (B) model 2 with parameters $K^* = 10$, $n = 450$.

**Table 6 Simulation running time.**

| $(n, K)$ | (100, 10) | (1,000, 10) | (1,000, 100) | (5,000, 10) | (5,000, 100) | (10,000, 500) |
|---|---|---|---|---|---|---|
| $T_C$ | 0.06 | 0.7 | 4.4 | 6.7 | 149 | 2,696 |
| $T_E$ | 0.08 | 0.1 | 0.2 | 0.6 | 1.9 | 11.6 |

is seen from the table that for large problems ($n = 10,000$, $K = 500$), the running time of EB estimation step was less than 1% of the runtime of spectral clustering.

## Real data examples

In this section, we apply our empirical Bayes method on two real-world networks. For these networks, we do not have the underlying connectivity matrix as the ground truth, which makes it difficult to evaluate estimation accuracy. However, for a network with known node labels that indicate their community memberships (the "ground truth"), the true partition $Z_{\text{true}}$ of the vertices is given. Thus, we will develop accuracy metrics based on $Z_{\text{true}}$ to compare different methods.

For real data, the assumption of the regular stochastic block model (2) may be restrictive. A commonly used model is the degree-corrected stochastic block model (DCSBM) (*Karrer & Newman, 2011*) that uses a Poisson distribution to model the number of edges across blocks and takes within-community degree heterogeneity into consideration. Some methods have been developed to compare the goodness of fit of different types of SBMs to real world networks. *Yan et al. (2012)* has proposed a method to select models for DCSBM, which is essentially a hypothesis test against the null model of a regular SBM. The method calculates a test statistic from node degrees and their labels, and compares the value of the statistic to a Gaussian distribution to obtain a $p$-value under the null SBM. We used this method to test whether the regular SBM is a good model for the two real-world networks.

### *Political blogs*

First we consider the French political blogosphere network from *Latouche, Birmelé & Ambroise (2011)*. The network is made of 196 vertices connected by 2,864 edges. It was built from a single day snapshot of political blogs automatically extracted on October 14th,

2006 and manually classified by the "Observatoire Presidentiel" project (*Zanghi, Ambroise & Miele, 2008*). In this network, nodes correspond to hostnames and there is an edge between two nodes if there is a known hyperlink from one hostname to the other. The four main political parties that are present in the data set are the UMP (french republican), liberal party (supporters of economic-liberalism), UDF (moderate party), and PS (french democrat). However, in the dataset annotated by *Latouche, Birmelé & Ambroise (2011)* there are $K^* = 11$ different node labels in total, since they considered analysts as well as subgroups of the parties. The test statistic by the method in *Yan et al. (2012)* yielded a *p*-value of 0.08 according to the bootstrap distribution suggested by the authors, which indicates that the regular SBM is a fair representation of this network compared to DCSBM.

Given the known community memberships, we constructed a connectivity matrix $\Theta^* = (\theta^*_{ab})_{K^* \times K^*}$ with entries

$$\theta^*_{ab} = X^B_{ab}/n_{ab}, \qquad a, b \in \{1, \ldots, K^*\}, \tag{24}$$

where $X^B_{ab}$ is the number of edges observed in block $(a, b)$, $n_{ab} = |Z^{-1}_{true}(a)| \cdot |Z^{-1}_{true}(b)|$ for $a \neq b$ and $n_{aa} = |Z^{-1}_{true}(a)| \cdot (|Z^{-1}_{true}(a)| - 1)/2$, and $K^*$ is the true number of communities. Then the MSE (18) between an estimate $\widehat{\Theta}(K)$ and $\Theta^*$ (24) were used as an accuracy metric to compare estimated connectivity matrices, where $K$ is the input number of clusters.

We also used test data likelihood as another comparison metric. We randomly sampled 70% of the nodes, denoted by $V_o$, as observed training data, and estimated a connectivity matrix $\widehat{\Theta} = (\hat{\theta}_{ij})_{K^* \times K^*}$ from their edge connections and true memberships. Denote by $V_t$ the test data nodes not used in the estimation. Recall that $X_{ij}$ is the $(i, j)$th element in the adjacency matrix of the network. Then test data likelihood $\mathscr{L}_{test}$ was calculated according to (2) given the $\widehat{\Theta}$ estimated by a method,

$$\mathscr{L}_{test} = \prod_{i \in Vo, j \in Vt} \hat{\theta}^{X_{ij}}_{z_i z_j} \left(1 - \hat{\theta}_{z_i z_j}\right)^{1-X_{ij}} \times \prod_{k < j \in Vt} \hat{\theta}^{X_{jk}}_{z_j z_k} \left(1 - \hat{\theta}_{z_j z_k}\right)^{1-X_{jk}}, \tag{25}$$

where $z_i, z_j, z_k$ are the known ground truth labels (ground truth) of the nodes. Note that $X_{ij} \in \{0, 1\}$ is the edge connection between a vertex $i$ in the training data and a vertex $j$ in the test data, while $X_{jk}$ is the edge connection between two vertices $j$ and $k$ in the test data. We repeated this procedure 100 times independently to find the distribution of test data likelihood $\mathscr{L}_{test}$ across random sample splitting of the $n$ nodes into $V_o$ and $V_t$.

We applied VBEM to detect communities with an input number of clusters $K = 1, 2, \ldots, 20$. The MSE ratios of EB over the other two competing methods were calculated and plotted against $K$ in Fig. 8A. It is clear that EB achieved smaller MSE than the other two methods for all values of $K$. When $K$ was close to or greater than $K^* = 11$, EB provided more accurate estimates than both MLE and VBEM with smaller MSEs. Figure 8B shows the box-plot of test data log-likelihood values across 100 random sample splitting. From the box-plots, we see that the test data likelihood of EB was significantly higher than the other two estimates. These comparisons confirm that EB estimates were more accurate than the other two competing methods in terms of both metrics. In terms of model

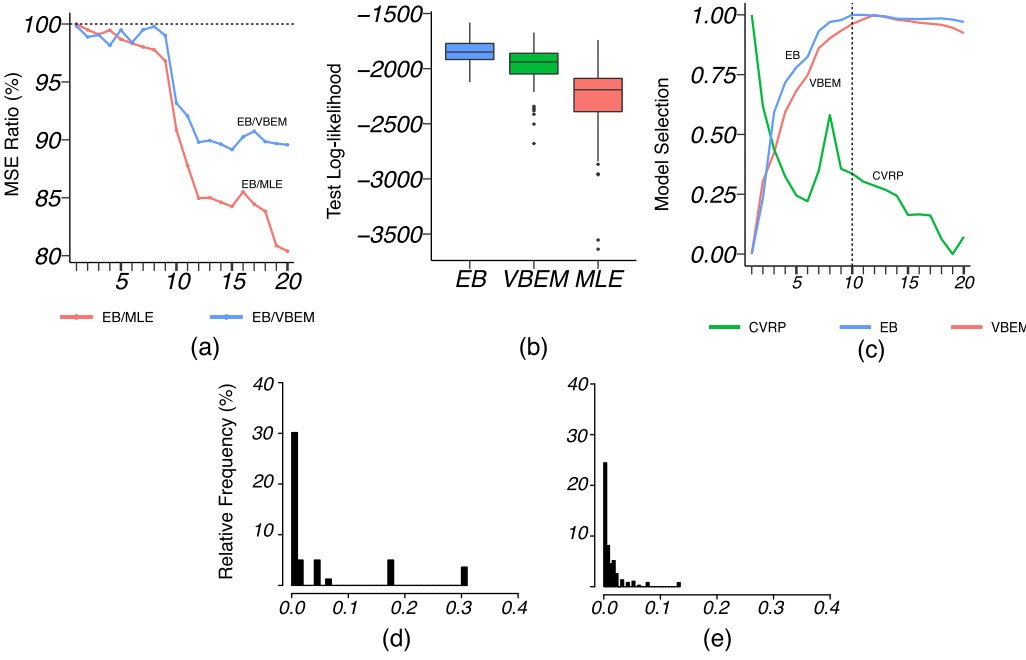

**Figure 8 Results for French blogsphere network analysis.** (A) The ratio of MSE of EB estimate over that of MLE and VBEM for different values of $K$. (B) Box-plot of 100 test data log-likelihood values for each method. (C) Model selection criteria against input values of $K$, with dashed line indicating the best number of clusters $\hat{K}$ by EB. (D and E) Histograms of (D) diagonal and (E) off-diagonal shrinkage values $\eta_{ab}$ of EB estimate at $K = K^* = 11$.

selection, Fig. 8C plots the three model selection criteria, $\mathscr{J}_{CVRP}$, $\mathscr{J}_{VBEM}$, $\mathscr{J}_{EB}$, over the input range of $K$. All three model selection criteria have been standardized to [0, 1] with a higher value indicating a better model, such that the best model is selected by the maximizer of each criterion. Accordingly, CVRP, VBEM and EB estimated $\hat{K} = 1$, 12 and 10, respectively, while the true $K^* = 11$. The $\hat{K}$ by VBEM and EB were both reasonably close to the ground-truth, while CVRP did not work well in this case. Figures 8D and 8E show the distributions of the shrinkage values $\eta_{ab}$ at $K = K^*$. We see that the diagonal blocks had higher shrinkage. Around 70% of the $\eta_{ab}$'s were around 0, which means that most blocks had a similar estimate to the MLE, while a few blocks with large $\eta_{ab}$ borrowed information from shrinkage and increased the estimation accuracy.

## Email network

The Email-Eu-core network (Eucore) is a directed network generated using email data from a large European institute, consisting of incoming and outgoing communications between members of the institute from 42 departments. *Leskovec & Krevl (2014)* organized the data and labeled which department each individual node belongs to, *i.e.*, the "ground-truth" community memberships. The network has $n = 1,005$ nodes and 25,571 directed edges, which we converted to undirected ones by removing their orientations. Although the test of *Yan et al. (2012)* suggested rejection of the hypothesis that a regular SBM is the true underlying model, our results on this network still show the

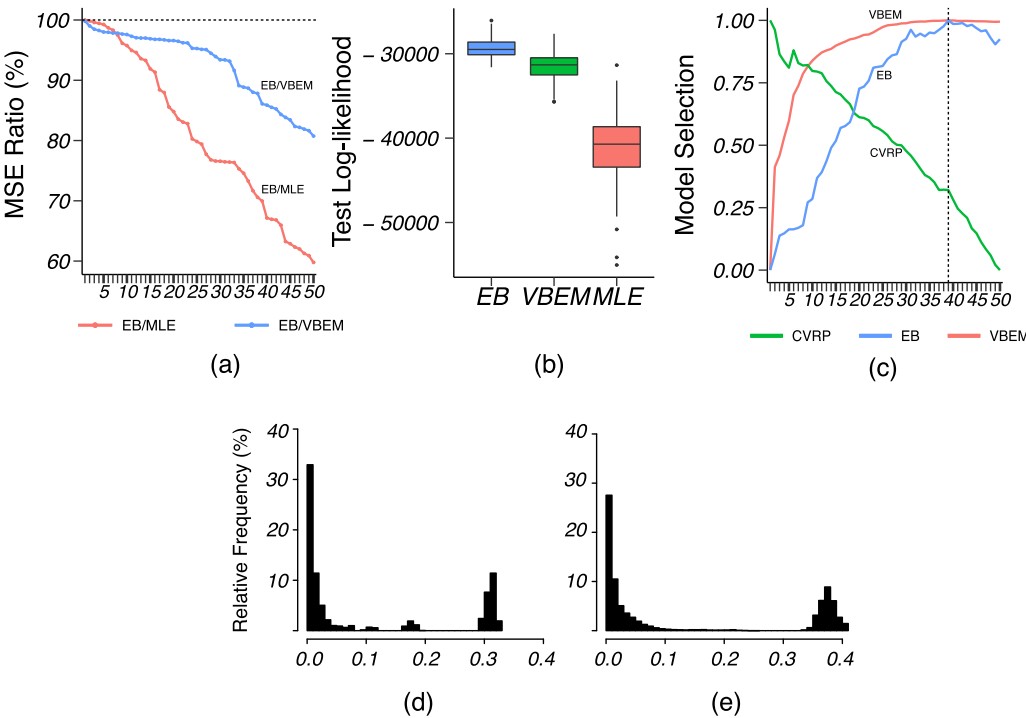

**Figure 9 Results for Email-Eu-core network analysis.** (A) The ratio of MSE of EB estimate over that of MLE and VBEM for different values of $K$. (B) Box-plot of 100 test data log-likelihood values for each method. (C) Model selection criteria against input values of $K$, with dashed line indicating the estimated $\hat{K}$ by EB. (D and E) Histograms of (D) diagonal and (E) off-diagonal shrinkage values $\eta_{ab}$ of EB estimate at $K = K^* = 42$.

improvement brought by EB assuming a regular SBM. We leave the generalization of our EB estimate to DCSBM as a future direction.

We applied VBEM to detect communities with an input number of clusters $K = 1$, $2, \ldots, 50$. The MSE ratios of EB over the other two competing methods were calculated and plotted against $K$ in Fig. 9A. Similarly, EB achieved smaller MSE than the other two methods for all values of $K$. The MSE ratios ranged from 60% to 90%. When the input number of communities $K$ was close to or greater than $K^* = 42$, the improvement of EB over the competing methods became more substantial. Figure 9B shows higher test data likelihood of EB than the other two estimates. Figure 9C shows the values of three model selection criteria for $k \in \{1, \ldots, 50\}$. The three methods, CVRP, VBEM and EB, gave estimates $\hat{K} = 1$, 39 and 39, respectively. The $\hat{K}$ by VBEM and EB were both reasonably close to the ground-truth of $K^* = 42$, while the performance of CVRP was much worse on this dataset. Moreover, $\mathscr{I}_{VBEM}$ is relatively flat around the estimated $\hat{K}$, while the curve of $\mathscr{I}_{VBEM}$ shows a higher sensitivity. From the distributions of the shrinkage values $\eta_{ab}$ in (d) and (e), we see $\eta \geq 0.3$ for a good number of diagonal and off-diagonal blocks, which led to substantial shrinkage and better performance than the MLE.

## DISCUSSION

We first briefly summarize this article and then discuss some limitations of this work and potential generalizations in future work.

### Summary

In this article, we developed an empirical Bayes estimate for the probabilities of edge connections between communities in a network. While empirical Bayes (EB) under a hierarchical model is a well-established method, its application to SBMs is very limited before our work. Our method is a natural fit to the SBM and the idea is generally applicable to different community detection methods. It does not require complicated algorithms or heavy computation, yet can effectively improve the estimation accuracy of model parameters. For the large volume of published community detection or network clustering algorithms, our parameter estimation method can be adopted as a superior alternative after the node clustering step. SBM approximation to graphons could result in a large number of blocks, for which case the EB often shows substantial advantage over the MLE, and this was a key motivation for our generalization to graphon estimation. This also helps the development of a good model selection criterion based on the marginal likelihood.

Though shrinkage in empirical Bayes approach leads to more accurate estimate of the connectivity probabilities, the improvement depends on the variability of the underlying connectivity matrix or graphon function. Typically, a higher variance reduces its improvement relative to the MLE. Therefore, for some graphon functions with high volatility, EB cannot guarantee a better estimate, but from our simulation results, EB estimate and MLE are usually comparable for such cases. A main reason for this observation is that EB estimate uses a very small number of hyperparameters, which effectively reduces the model complexity *via* shrinkage and greatly minimizes the risk of overfitting the data.

In our experiments, we compared the model estimation accuracy by the mean squared error, which is a gold standard criterion to evaluate parameter estimation. However, several other metrics, such as the KL-divergence of the estimated graphon function to the truth, deviation of the estimated number of motifs in the graph to the true value, and divergence of degree distributions, can also be considered. For the application on real data, the goodness of fit of SBM or graphon model to the datasets may be compared to more existing network modeling methods in addition to DCSBM. A decent fit of the SBM and/or graphon to these datasets will further demonstrate the usefulness of our method in a more convincing way.

### Future work

We put a beta conjugate prior on connection probability $\Theta$, and the estimates of the hyperparameters $(\alpha_d, \beta_d)_{d=\{0,1\}}$ are always positive. Thus, when a true connectivity $\theta_{ab} = 0$ for some block $(a, b)$, which is likely to happen in sparse networks, our hierarchical model introduces bias to the estimate of $\theta_{ab}$ by Eq. (6). However, since the empirical Bayes estimator is pooling data in all the blocks, the overall accuracy measured by MSE is still expected to be higher. To alleviate this bias, we may consider a proportion $\gamma$ of zero

connectivity blocks and only apply shrinkage across blocks with a nonzero connectivity parameter.

We have focused on parameter estimation for binary and assortative stochastic block models and graphons. For some real-world applications, a regular SBM may not be the most appropriate model, and degree corrected SBM mentioned above is usually a better choice, in which the edge variable $A_{ij}$ between two nodes $i, j$ is modeled as

$$A_{ij}|z_i, z_j \sim \mathrm{Poisson}(\theta_{z_i z_j}\omega_i\omega_j), \tag{26}$$

where $z_i$ and $z_j$ are the node community labels. The node-specific parameter $\omega_i$ scales the number of connections to allow different expected degrees. The idea of empirical Bayes can be generalized for this model: After community labels are determined by a graph clustering algorithm, the MLE of $\omega_i$, which only involves degree distributions and community labels, can be calculated. After we plug in these MLEs, we can construct a hierarchical model for the parameters $\theta_{ab}$ with a conjugate Gamma prior, which leads to a similar empirical Bayes estimator *via* shrinkage across multiple blocks.

Furthermore, the idea can be generalized to more sophisticated random graph models, such as SBM with mixed memberships (*Airoldi et al., 2008*), SBM with weighted edges (*Aicher, Jacobs & Clauset, 2015*), and bipartite SBM (*Larremore, Clauset & Jacobs, 2014*) *etc*. While most of the related works focus on graph clustering, our empirical Bayes method can be applied after clustering to improve the estimation accuracy and to identify a proper number of blocks for these models.

### Funding
This work was supported by NSF grant DMS-1952929. There was no additional external funding received for this study. The funders had no role in study design, data collection and analysis, decision to publish, or preparation of the manuscript.

### Grant Disclosures
The following grant information was disclosed by the authors:
NSF Grant: DMS-1952929.

### Competing Interests
The authors declare that they have no competing interests.

### Author Contributions
- Zhanhao Peng conceived and designed the experiments, performed the experiments, analyzed the data, performed the computation work, prepared figures and/or tables, authored or reviewed drafts of the article, and approved the final draft.
- Qing Zhou conceived and designed the experiments, authored or reviewed drafts of the article, and approved the final draft.

## Data Availability

The data is available at GitHub: https://github.com/chandler96/EBgraph.

## Supplemental Information

Supplemental information for this article can be found online at http://dx.doi.org/10.7717/peerj-cs.1006#supplemental-information.

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
