# Peer review of "An empirical Bayes approach to stochastic blockmodels and graphons: shrinkage estimation and model selection"

_PeerJ Computer Science, doi:10.7717/peerj-cs.1006_

## Round 0.1 · original submission · Major Revisions

· Academic Editor

Major Revisions

Experts' careful review of this manuscript has found that the paper needs major revisions before being accepted for publication in this journal. Some of the main concerns pointed out by reviewers are summarised as follows:

1) The literature cited in the paper is relevant, but it does not reflect the breadth of research done on the topic.
2) The description of methods used for simulations and data analysis should be clarified with the exact steps mentioned clearly.
3) Further work and transparent reporting should help confirm the paper's validity and contribute to broader its interest.

Reviewer 1 ·

Basic reporting

The papers appears to meet all the basic reporting requirements of the journal. Regarding the literature review, I only have one minor concern (see bullet point #1 in my list in the section ‘Additional comments’).

Experimental design

The papers appears to meet all the experimental design requirements of the journal.

Validity of the findings

The papers appears to meet all the requirements of the journal around validity of the findings.

Additional comments

The paper proposes a simple yet effective empirical Bayes estimator for the block connectivity matrix in stochastic blockmodel, conditional on an estimate of the community structure. The authors also discuss implications of their work in terms of graphon estimation, using the stochastic blockmodel approximation to graphons. The paper is well presented and the methodologies clearly explained. The methods presented in this article are particularly relevant for the communities interested in network analysis and community detection, therefore I believe the paper would fit well within the aims and scope of the journal. Some of my main concerns are summarised below:

1. The problem of correctly selecting the number of communities K is mentioned, but existing methods for estimation of K are not discussed. I believe a brief literature review on criteria for model selection should be included, considering that this is one of the central aspects of the work. For example, in the context of spectral graph clustering, a good reference for this would be Yang et al. (JCGS, 2021).
2. Page 4, before equation (12) — A Jeffreys prior is chosen for the partition probabilities, but no discussion is provided on the reasoning behind this choice and potential implications. What is the effect of the prior parameters \tau on the performance of the estimation procedure for K? It would be good to comment on this (maybe examining limit cases such as \tau to 0 or to infinity — or via simulations).
3. The simulations presented in the section ‘Results / Simulation’ (page 7 onwards) are fairly simple: the simulated networks are strongly assortative and the block connectivity probabilities are highly homogeneous. It is reasonable and expected to see a shrinkage estimator to have a better performance under this scenario. It would be interesting to see how the performance changes when heterogeneous block connectivity probabilities are used (maybe adding such additional simulations to the supplementary material).
4. Lines 182 and 188, page 8 — The two simulation scenarios are referred to as ‘dense’ and ‘sparse’. Commonly, sparsity for graphs is defined as the number of nodes grows to infinity and the number of edges grows sub-quadratically (see e.g. Caron and Fox, JRSSB, 2017). Here, the term ‘sparse’ appears to be used instead to mean that the setting for model 2 is just sparser than model 1. I believe that this should be clarified to avoid confusion for the readers. It might be appropriate to simply report the expected edge density under the two parameter settings, to better appreciate the difference between the two simulations.
5. The ‘Real data examples’ section on page 15 onwards assumes that the node labels available for the email network and political blogs network are actually representative of network connectivity. Often in real world networks, the assumption of a stochastic blockmodel is limiting, since it assumes extremely simple IID binomial within-community degree distributions — degree-corrected stochastic blockmodel (Karrer and Newman, 2011) might therefore be appropriate in the presence of within-community degree heterogeneity. Since the labels are used to assess the effectiveness of the proposed methodologies, would it be possible to evaluate (maybe via hypothesis testing) whether the blocks obtained from the available labels are representative of a stochastic blockmodel? In my opinion, this would make the real data analysis more convincing.
6. The choice of possible values of K for the email network appears to be only between 30 and 50. Many times in practice much smaller numbers of communities are sought. It would be interesting to see the performance of the models also for K starting from 1 as in the political blogs example, if possible.

Minor details:
The hyperreferences in the PDF document are not displayed correctly — see e.g. Figure ?? on page 5, line 130; (Section ) on page 7, lines 175-176.

References:
- Caron, F. and Fox, E.B. (2017), Sparse graphs using exchangeable random measures. J. R. Stat. Soc. B, 79: 1295-1366. https://doi.org/10.1111/rssb.12233
- Congyuan Yang, Carey E. Priebe, Youngser Park & David J. Marchette (2021) Simultaneous Dimensionality and Complexity Model Selection for Spectral Graph Clustering, Journal of Computational and Graphical Statistics, 30:2, 422-441, DOI: 10.1080/10618600.2020.1824870
- B Karrer, MEJ Newman (2011), Stochastic blockmodels and community structure in networks, Physical Review E 83 (1), 016107

Reviewer 2 ·

Basic reporting

The article is mostly well-written, however there are a few sentences which are not clear (please see specific comments below) and improvements can be made in the section on future work. The literature cited is relevant however it does not reflect the breadth of research done on the topic. Thus, I would recommend the authors to find the latest articles on this topic without necessarily restricting to Bayesian techniques. More can be done to provide a deeper context to the problem. A simple example illustrating the requirement for shrinkage with existing graphon estimation techniques say would be helpful. The paper is structured well and figures, tables etc are clear.

Experimental design

The research question is well defined and meaningful, however, as mentioned above, a specific example would be helpful in providing context and understanding the relevance of the shrinkage problem.
Investigations have been performed via simulations and two data sets (please see specific comments regarding this). The description of methods used for simulations and data analysis should be further clarified with the exact steps mentioned clearly.

Validity of the findings

Benefit to literature is clearly stated but is not fully evident from existing analyses and presentation. Some further work and a clear reporting should help confirm the validity of the findings and also make the contribution of wider interest.

Additional comments

Specific comments:

p.1. line 40: clarify symmetry/undirected networks
p.2, line 74: `Empirical Bayes method is usually seen to have better performance when estimating many similar and variable quantities' :
-What does 'similar' and 'variable' mean exactly and why is this the case briefly?
- p.2, line 76: How `small' is the network size that you refer to here?
-p.5, line 130: missing figure number (??). Please check carefully throughout the document for referencing.
-p.7, line 175-176: labels of sections missing
-p.15, line 267-8: need rephrasing it seems. Not clear.
-p.2, around eqn 2.: is it assumed that n nodes are divided into K blocks, each allowed to be of a different size? If yes, please include it in the description here.
-p.5, line 130: Can there be scenarios where L-BFGS may not succeed in finding the global maxim? Please comment on this in the paper.
-p.6: what is the motivation for using ICL (and e.g. not Bayes factors)?
p.11: simulations on graphon estimation. What does the graphon model look like? Can a heatmap be included? Results using another graphon example with a different structure would be useful (google scholar results for `graphon estimation' brings up latest articles, many of which report simulation comparisons for a variety of graphon structures).
p.11, line 247: the performance of your method over competing methods is reported to be significant in the sparse case. Is this expected? If so, why?


- With only the K specified, how are the sizes of different clusters determined ( if they are assumed to be unequal in general)?
- In practice multiple labelings of nodes (i.e. different Z's) may be available from different clustering methods. What is the best way to proceed given this?
- How sensitive is your approach to different Z's in practice? This is not clear from the simulations.
- How may the range of K determined in practice? For example, why is K chosen to be at least 30 and at most 50 in the first data example?
- What are the labels z's in eqn (25). It is mentioned that these are 'known labels of the nodes' - what does this mean exactly? Are these the ground truth labels or labels estimated from an existing graph clustering algorithm?


My main concern is the simulation and data analyses reported to validate the findings. The MSEs can give a different picture and sometimes be deceptive. I would recommend plotting heatmaps of the estimates from each method directly for a visual confirmation that all methods are working as expected. Further, I understand that this is a Bayesian technique and thus it is fair to include comparisons with other Bayesian methods, however given the vast literature on this topic, it is important that the proposed method is compared with atleast one other non-Bayesian approach (e.g. network histogram of Olhede & Wolfe, 2014 is commonly used).

Secondly, why is it fair to use the department-wise labels as the ground truth in the first data example? There may be clusters based on multiple attributes or a combination of them and it is difficult to define ground truth based on node attributes in general. Is there a justification for this approach in this particular case/data set? One of the commonly used methods is to compare the mean squared prediction errors (as in recent graphon estimation papers in statistics for example).

It would be useful to see the values that the shrinkage parameter assumes to get insights on what determines estimation of connection probabilities in the two data sets. It would be useful to report the estimated shrinkage \eta's if possible.

---

## Round 0.2 · accepted · Accept

· Academic Editor

Accept

Your manuscript "An empirical Bayes approach to stochastic blockmodels and graphons: shrinkage estimation and model selection" has been re-reviewed. I am happy to inform you that it has been accepted for publication. The revised version of the manuscript addressed all the concerns that the reviewers expressed in their previous review.

Reviewer 1 ·

Basic reporting

The papers appears to meet all the basic reporting requirements of the journal.

Experimental design

The papers appears to meet all the experimental design requirements of the journal.

Validity of the findings

The papers appears to meet all the basic validity of the findings requirements of the journal.

Additional comments

The revised version of the manuscript addressed the concerns that I expressed in my previous review. I thank the authors for implementing the suggested edits.